# Prospective associations between coffee consumption and psychological well-being

Farah Qureshi [1,2]*, Meir Stampfer[3,4], Laura D. Kubzansky[1,2], Claudia Trudel-Fitzgerald[1,2¤a¤b]

**1** Department of Social and Behavioral Sciences, Harvard T.H. Chan School of Public Health, Boston, MA, United States of America, **2** Lee Kum Sheung Center for Health and Happiness, Harvard T.H. Chan School of Public Health, Boston, MA, United States of America, **3** Department of Epidemiology, Harvard T.H. Chan School of Public Health, Boston, MA, United States of America, **4** Channing Division of Network Medicine, Department of Medicine, Brigham and Women's Hospital and Harvard Medical School, Boston, MA, United States of America

¤a Current address: Department of Psychology, Université du Québec à Trois-Rivières, Trois-Rivières, Québec, Canada
¤b Current address: Research Center of Institut Universitaire en Santé Mentale de Montréal, Montréal, Québec, Canada
* fqureshi@mail.harvard.edu

**Data Availability Statement:** The data that support the findings of this study are available from the Channing Division of Network Medicine at Brigham and Women's Hospital, but restrictions apply to the

## Abstract

### Objective

Prior work indicates a robust relationship between coffee consumption and lower depression risk, yet no research has examined links with psychological well-being (e.g., happiness, optimism). This study tested whether coffee intake is prospectively associated with greater psychological well-being over time. Secondarily, associations in the reverse direction were also examined to determine whether initial levels of psychological well-being were related to subsequent coffee consumption.

### Methods

Among women in the Nurses' Health Study, coffee consumption was examined in 1990 and 2002 in relation to sustained levels of happiness reported across multiple assessments from 1992–2000 (N = 44,449) and sustained levels of optimism assessed from 2004–2012 (N = 36,729). Associations were tested using generalized estimating equations with a Poisson distribution adjusted for various relevant covariates. Bidirectional relationships were evaluated in secondary analyses of baseline happiness (1992) and optimism (2004) with sustained moderate coffee consumption across multiple assessments through 2010.

### Results

Compared to minimal coffee consumption levels (<1 cup/day), moderate consumption (1–3 cups/day) was unrelated to happiness, whereas heavy consumption (≥4 cups/day) was associated with a 3% lower likelihood of sustained happiness (relative risk, RR = 0.97, 95% CI = 0.95–0.99). Only moderate coffee consumption was weakly associated with a greater likelihood of sustained optimism (RR$_{1-3cups/day}$ = 1.03, 95% CI = 1.00–1.06). Secondary

availability of these data. Data may be available to access upon reasonable request and with the permission of the Channing Division of Network Medicine at Brigham and Women's Hospital (contact via nhsaccess@channing.harvard.edu).

**Funding:** The Nurses' Health Study was funded by a grant from the National Institutes of Health (NIH) (grant number UM1 CA186107). This study also received funding from the National Institutes of Health training (grant T32 CA 009001) awarded to FQ, and through an unrestricted gift from the United Nations Sustainable Development Solutions Network, which is supported in part by a gift from the Illy Foundation, awarded to MS. The Lee Kum Sheung Center for Health and Happiness provided support in the form of salary for CTF. The content of this manuscript is the sole responsibility of the authors and does not necessarily represent the official views of the funding agencies, which did not review the manuscript prior to submission for publication.

**Competing interests:** The authors declare that they have no conflicts of interest.

analyses showed high levels of optimism but not happiness levels were modestly associated with increased likelihood of sustained moderate coffee intake ($RR_{optimism}$ = 1.06, 95% CI = 1.02–1.10; $RR_{happiness}$ = 1.01, 95% CI = 0.99–1.04).

## Conclusions

Associations between psychological well-being and coffee consumption over up to two decades were largely null or weak. Although coffee consumption may protect individuals against depression over time, it may have limited impact on facets of psychological well-being.

## Introduction

Growing evidence suggests psychological well-being (e.g., optimism, happiness, life purpose) is associated with reduced likelihood of poor health over the life course [1–4]. This lower risk may be due in part to protective associations with lifestyle factors involved in health maintenance and decline, such as physical activity, diet, and tobacco use [3, 5, 6]. For example, recent work among over 35,000 women in the Nurses' Health Study (NHS) found that higher levels of happiness and optimism were each associated with a 39% to 40% greater likelihood of maintaining a healthy lifestyle (defined using an index of physical activity, body mass index, diet, alcohol use, and tobacco consumption) over 10–22 years, and associations were robust to adjustment for sociodemographic factors and depression [7]. Inversely, understanding factors that promote psychological well-being may also provide important insights into strategies for improving both mental and physical health. In prior work, it was shown that that adopting a healthy lifestyle was related to an increased likelihood of experiencing sustained high levels of happiness and optimism over time (relative risk, $RR_{happiness}$ = 1.11, 95% confidence interval (CI) = 1.10–1.12; $RR_{optimism}$ = 1.26, 95% CI = 1.22–1.31), suggesting that health behaviors may play a role in promoting psychological well-being [7].

Another health behavior associated with potential psychological benefits is coffee consumption [8]. Numerous studies have demonstrated associations between higher levels of coffee intake and a lower risk of suicide and depression [9–12]. In a pooled analysis of three large-scale US-based cohorts of middle-aged men and women (including NHS) [13], each additional two cups/day of coffee consumed was associated with a 25% lower risk of suicide (pooled multivariable RR = 0.75, 95% CI = 0.63, 0.90). A study conducted among over 50,000 NHS women similarly evaluated coffee intake in relation to incident depression over a 10-year period [11]. Compared to the lowest coffee consumption level ($\leq$1 cup/week), women who drank 2–3 cups/day and $\geq$4 cups/day experienced a 15% and 20% lower risk of becoming depressed, respectively, after controlling for sociodemographic factors, health behaviors, comorbidities, and social engagement (e.g., retirement status, community engagement; fully adjusted $RR_{2-3\ cups}$ = 0.85; 95% CI = 0.75–0.95; fully adjusted $RR_{\geq4\ cups}$ = 0.80; 95% CI = 0.64–0.99). In a recent meta-analysis [14], pooled associations estimated across 327,697 men and women in four prospective and two cross-sectional investigations (including the NHS study described above) found that those who reported the highest versus lowest levels of coffee intake were 24% less likely to be depressed ($RR_{pooled}$ = 0.76, 95% CI = 0.64–0.91). When considering only prospective findings that evaluated baseline coffee consumption in relation to future risk of depression over up to 17.5 years of follow-up, the pooled effect estimate was attenuated but still sizeable (RR = 0.88, 95% CI = 0.79–0.99).

Although prior findings indicate that coffee may be psychologically protective with respect to suicide and depression, no research has evaluated whether coffee consumption may contribute to experiencing psychological well-being over time. Because the absence of psychological distress does not necessarily denote the presence of psychological well-being (i.e., individuals who are not depressed may not experience high levels of well-being) [4, 15], such an investigation is warranted. Furthermore, preliminary evidence suggests that this association may be more relevant for women given potential sex differences in how coffee is metabolized, with women possibly experiencing slower systemic clearance than men [16]. The primary goal of this study was to examine the relationship between coffee consumption and two commonly measured facets of psychological well-being–happiness and optimism–over 10 years of follow-up among women enrolled in NHS. Considering previous findings with depression in the same cohort [14], it was hypothesized that higher levels of coffee consumption at baseline would be associated with increased likelihood of sustaining high levels of psychological well-being over time, even after controlling for a range of relevant confounders, including sociodemographic factors, other health-related biobehavioral factors, and social engagement. Further adjustment accounted for depression to determine whether associations of coffee with happiness and optimism, respectively, are noted over and above any depressive symptoms. Bidirectional associations were also explored, namely whether initial levels of these facets of psychological well-being were prospectively associated with subsequent coffee consumption over up to 20 years. Following prior results indicating bidirectional associations between psychological well-being and healthy lifestyle factors [7], it was posited that greater happiness and optimism at baseline would be associated with a greater likelihood of sustaining moderate levels of coffee consumption over time. In secondary analyses, levels of decaffeinated coffee, nonherbal caffeinated tea, and total caffeine intake were also evaluated to determine whether associations with facets of psychological well-being are attributable to the caffeine content in coffee.

## Methods

### Participants

The Nurses' Health Study (NHS) is an ongoing U.S.-based longitudinal cohort that enrolled 121,700 female registered nurses aged 30–55 in 1976 [17], and has since tracked their lifestyle, medical history, and newly diagnosed conditions through biennial questionnaires (>85% follow-up) [18]. The present study investigated happiness and optimism, which were assessed for the first time at different questionnaire cycles in the NHS, necessitating the construction of two distinct analytic samples. For happiness analyses, coffee intake in 1990 was examined in relation to repeated measures of happiness over three follow-up assessments from 1992 to 2000. For optimism analyses, coffee intake in 2002 was examined in relation to repeated measures of optimism over three follow-up assessments from 2004 to 2012. For a timeline illustrating the sequence of exposure and outcome assessments, see S1 Fig in S1 File. The study protocol was approved by the institutional review board for Brigham and Women's Hospital and the Harvard T.H. Chan School of Public Health.

A flowchart illustrating the composition of each of the two samples used for the primary analyses is provided in Fig 1. Women were excluded from our study samples if they had missing data on average daily coffee consumption at each analytic baseline (i.e., happiness analyses: 1990; optimism analyses: 2002). Following prior work examining relationships between psychological well-being and healthy lifestyle [7], participants were also excluded if they reported a major chronic condition (i.e., prevalent cancer excluding non-melanoma skin cancer, diabetes, myocardial infarction, or angina) at the study onset since participants'

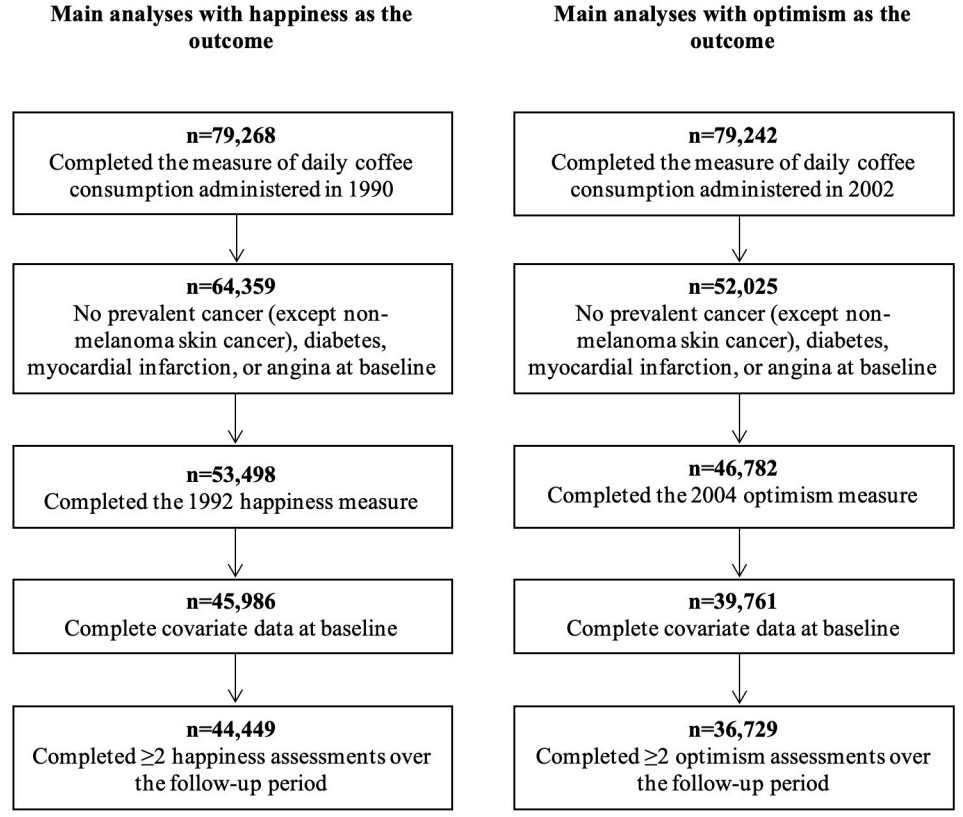

**Main analyses with happiness as the outcome**

**n=79,268**
Completed the measure of daily coffee consumption administered in 1990

↓

**n=64,359**
No prevalent cancer (except non-melanoma skin cancer), diabetes, myocardial infarction, or angina at baseline

↓

**n=53,498**
Completed the 1992 happiness measure

↓

**n=45,986**
Complete covariate data at baseline

↓

**n=44,449**
Completed ≥2 happiness assessments over the follow-up period

**Main analyses with optimism as the outcome**

**n=79,242**
Completed the measure of daily coffee consumption administered in 2002

↓

**n=52,025**
No prevalent cancer (except non-melanoma skin cancer), diabetes, myocardial infarction, or angina at baseline

↓

**n=46,782**
Completed the 2004 optimism measure

↓

**n=39,761**
Complete covariate data at baseline

↓

**n=36,729**
Completed ≥2 optimism assessments over the follow-up period

**Fig 1. Flowchart of Nurses' Health Study participants included in the final analytic samples.**

baseline health status might influence both their likelihood of consuming coffee [19] as well as their psychological well-being [20, 21]. Participants with missing data on baseline psychological well-being were also excluded, as well as those who only had data from one psychological well-being assessment over the follow-up period, and those with incomplete covariate information (see S1 Table in S1 File for the distribution of covariates among women who had complete and incomplete data at the baseline psychological well-being assessment). The final samples included 44,449 women for happiness analyses, and 36,729 women for optimism analyses. Although most included women had data on all psychological well-being assessments throughout the follow-up period, those who did not were somewhat older, less likely to have a graduate degree, be married or in a relationship, or socially integrated, and they were slightly more likely to be depressed (see S2 Table in S1 File for the distribution of covariates among women who completed all versus some of the psychological well-being assessments).

Separate analytic sub-samples were also derived for secondary analyses examining the association between baseline psychological well-being and sustained coffee intake over time. For happiness analyses, happiness in 1992 was examined with repeated measures of coffee intake over 5 follow-up assessments from 1994 to 2010. For optimism analyses, optimism in 2004 was examined in relation to repeated measures of coffee intake over two follow-up assessments from 2006 to 2010. For these secondary analyses, sub-samples were derived from the primary analytic samples by further excluding women who only had data from one coffee assessment over the follow-up period ($N_{happiness\ analyses}$ = 43,489; $N_{optimism\ analyses}$ = 31,441).

## Measures

**Coffee consumption.** Women's consumption of caffeinated coffee was measured via a self-reported food frequency questionnaire that assessed usual intake levels over the past 12 months [22]. Baseline measures were assessed in 1990 and 2002 for the happiness and optimism analytic samples, respectively, using 9 categories indicating the number of cups of coffee participants consumed over a given time period, ranging from <1 cup/week to ≥6 cups/day. Based on the frequency distribution in the analytic samples and evidence of a reduced risk of depression associated with drinking ≥1 cup/day [11], response options were collapsed into three levels of average consumption (minimal: <1 cup/day [reference], moderate: 1–3 cups/day, heavy: ≥4 cups/day). A continuous measure of coffee consumption frequency (standardized; per 1 standard deviation [SD]) was also considered. Subsequent coffee intake was queried every four years through 2010 (the latest available time point). These follow-up assessments were used in secondary analyses, in which we evaluated the association of baseline psychological well-being with the likelihood of maintaining moderate levels of coffee consumption (i.e., 1–3 cups/day), which is suggested to be optimal for health [23]. We defined sustained moderate coffee consumption (yes/no) as moderate intake reported in at least two assessments over the study period (not necessarily consecutive), including the baseline assessment.

Data on decaffeinated coffee, non-herbal caffeinated tea, and total caffeine intake were also available at each analytic baseline and used in sensitivity analyses. Decaffeinated coffee and caffeinated tea consumption were measured using the same levels as described above, whereas total caffeine was calculated based on participants' self-reported consumption of coffee, tea, soft drinks, and chocolate. Methods used to calculate total caffeine intake are based on food consumption data from the US Department of Agriculture and are described elsewhere [22]. Following prior work [11], caffeine consumption was categorized as minimal: <100 mg/day [reference], moderate: 100–549 mg/day, and heavy: ≥550 mg/day.

**Psychological well-being.** In 1992, 1996, and 2000, *happiness* was assessed with a single item included in the Medical Outcomes Study SF-36 [24]: "How much of the time during the last month have you been a happy person?" Responses ranged from 1 "all the time" to 5 "none of the time," and were reverse-coded so that a higher value indicated more happiness. Although single-item happiness or other psychological well-being measures are somewhat limited, they have predicted all-cause mortality [1], cardiometabolic disease risk [4], and health-related behaviors [7] in prior research. In the current study, happiness scores were modestly consistent across follow-up assessments (intra-class correlation coefficient [ICC = 0.44]).

In 2004, 2008, and 2012, *optimism* was measured using the validated 6-item Life Orientation Test-Revised [25], which asked women to rate the degree to which they agreed with statements describing a positive outlook on life (e.g., "In uncertain times, I usually expect the best"). Responses ranged from 1 "strongly disagree" to 5 "strongly agree." All responses were summed after reverse coding negatively worded items to create a total score ranging from 6 to 30, with higher scores indicating higher optimism levels. The stability of measures over time was reasonably high (ICC = 0.63), and Cronbach's alpha at each assessment indicated good internal consistency ($\alpha_{2000} = 0.78$; $\alpha_{2008} = 0.76$; $\alpha_{2012} = 0.76$).

Following prior work [7], three levels of each psychological well-being measure were created based on the distribution of scores in their respective sample at each assessment. Similar to findings in other cohorts [26], most women reported high levels of psychological well-being, therefore women were unequally divided across happiness levels (e.g., at the 1992 assessment: low = 14.5% [reference]; moderate = 18.6%; high = 66.9%). For optimism, women were more evenly divided (e.g., at the 2004 assessment: low = 32.0% [reference]; moderate = 30.8%; high = 37.3%). Participants were categorized as having sustained high happiness

or optimism (yes/no) if they had high levels at two or more assessments over the study period, including the baseline assessment.

**Covariates.** Potential confounders and other covariates were measured at each analytic baseline, unless otherwise specified, and included sociodemographic, behavioral, social, and psychological factors. Sociodemographics included age (continuous; years), race (White, non-White [reference]), education level (registered nurse [reference], bachelor, master's, doctorate; queried in 1992 only), and marital status (married/in a relationship, divorced/separated/widowed [reference]). Health-related behaviors encompassed physical activity, diet quality, smoking, alcohol consumption. Body mass index (BMI) was also considered. Prior work finds that these behaviors tend to cluster with each other and with BMI [27], and contribute to a lifestyle that has a multiplicative effect on the likelihood of mortality compared to individual habits [28]. Therefore these factors were aggregated into a lifestyle index (continuous; scores ranging from 0 to 5; details in S1 Text in S1 File) following procedures used in previous studies [7, 28–30] and consistent with cancer and cardiovascular prevention guidelines [31, 32]. Social integration was also considered since associations between coffee intake and psychological well-being may be confounded by social processes [14]. More specifically, higher levels of social integration may lead to greater levels of coffee consumption and psychological well-being, respectively. Social integration was measured via the Berkman-Syme Social Network Index [33], which defines levels of social integration (socially isolated [reference], moderately isolated, moderately integrated, socially integrated) based on four distinct dimensions of social networks: marital status; number of close relatives and close friends, separately; frequency of religious activities; and frequency of activities with community organizations (details in S2 Text in S1 File). Depression (yes, no [reference]) was used as an indicator of psychological distress and was defined as reporting either a physician-diagnosed depression or regular antidepressant use in the past two years.

## Statistical analyses

The distribution of study covariates by levels of coffee consumption at baseline (happiness analytic sample: 1990; optimism analytic sample: 2002) was described using means and proportions. The prevalence of sustained high levels of happiness and optimism as well as sustained moderate coffee consumption were calculated, and within-subject coefficients of variations (CVs) [34] were assessed to quantify the stability of psychological well-being and coffee intake, respectively, over the follow-up period. As described in detail below, a series of parallel analyses were then conducted to explore prospective associations between coffee consumption and happiness and optimism, respectively. First, baseline coffee consumption was examined in relation to sustained high levels of happiness and optimism. Next, secondary analyses investigated the association of baseline levels of happiness and optimism with sustained moderate levels of coffee consumption over time. In two distinct sets of sensitivity analyses, associations were tested with baseline levels of decaffeinated coffee, caffeinated tea, and total caffeine intake, and adjusting for baseline outcome levels.

**Primary analyses.** To account for correlated observations across repeated assessments, multivariable generalized estimating equations (GEE) were used to test whether baseline coffee intake was associated with sustained happiness and optimism levels over the study period (up to 2000 and 2012, respectively). Following recommendations for studies involving non-rare outcomes (i.e., those with a prevalence >10%), all GEE models used a Poisson distribution [35]. Associations between baseline coffee intake and sustained high happiness and sustained high optimism were each tested in a series of models that increasingly adjusted for covariates, including sociodemographic, lifestyle, and psychosocial factors. Model 1 (minimally adjusted)

controlled for age, race, education, and marital status. Model 2 additionally adjusted for participants' lifestyle scores, while Model 3 further adjusted for social integration levels. Model 4 (fully adjusted) additionally accounted for depression to verify whether associations of coffee intake with facets of psychological well-being are independent of psychological distress. Coffee consumption was examined as a 3-level measure (i.e., <1 cup/day [reference], 1–3 cups/day, ≥4 cups/day), with evaluation for a linear trend using the median value of each level. Associations were also tested using a continuous, standardized measure of coffee consumption frequency (per 1-SD).

**Secondary analyses.** To explore the possibility of bidirectionality, associations between baseline levels of psychological well-being (happiness analytic sample: 1992; optimism analytic sample: 2004) and sustained moderate coffee intake over the study period (up to 2010; see S1 Fig in S1 File) were tested using multivariable GEE models that accounted for covariates using a set of increasingly adjusted models with the same covariates as described in the primary analyses. Likewise, baseline facets of psychological well-being were included in separate models using either 3-level measures of happiness and optimism (i.e., low [reference], moderate, high), or continuous, standardized measures (per 1-SD).

**Sensitivity analyses.** In a first series of sensitivity analyses, data on decaffeinated coffee, caffeinated tea, and total caffeine intake were used to explore whether potential associations with facets of psychological well-being may be attributable to the caffeine content in coffee. A second series of sensitivity analyses aimed to mitigate potential bias induced by initial levels of the outcome influencing the experience or report of the exposure. To this end, primary and secondary analyses described above were re-evaluated while controlling for baseline levels of the relevant outcome, instead of including the baseline assessment in the outcome definition. For example, in primary models, coffee intake at baseline (1990) was examined in relation to sustained high happiness levels at follow-up (1996 and 2000) while controlling for initial happiness levels (1992). Similarly, in secondary models, associations between psychological well-being at baseline and sustained moderate coffee intake over the follow-up period were tested while considering initial levels of coffee intake (e.g., happiness in 1992 with sustained coffee intake from 1994 to 2010, while adjusting for coffee intake in 1990). All sensitivity analyses used GEE models with a Poisson distribution, except for those evaluating optimism's role in sustained coffee intake that controlled for baseline optimism, which used logistic regression with a Poisson distribution since only one assessment of the outcome was available (i.e., coffee intake in 2010).

**Inverse probability weights.** To address potential selection bias induced by differences noted when comparing women who completed all versus some of the follow-up assessments (see S2 Table in S1 File), person- and time-specific inverse probability weights were included in the models [36]. Specifically, the probability of participating at each time point was modeled based on the exposure and covariates of interest among each analytic sample, and a weight was created that corresponded to the inverse of the probability of participating. All analyses were conducted using SAS 9.4 with a two-sided $p$-value of 0.05.

## Results

### Baseline characteristics

The mean age of women at the 1990 happiness baseline was 56.0 years (SD = 7.1; range = 43–70) and most were White (98.3%), married (83.3%), reported registered nurse as the highest education level obtained (68.6%), and moderately-to-highly socially integrated (60.9%). The majority were not current smokers (83.3%) and had a healthy BMI (54.7%), but fewer engaged in favorable levels of physical activity (32.6%) and only 20% reported drinking on average 1 alcoholic drink/day. Characteristics were similar at the 2002 optimism baseline; however,

participants were older (mean age = 66.6; SD = 6.8; range = 55–82). When examining study characteristics by levels of baseline coffee consumption, heavy coffee drinkers (≥4 cups/day) were more likely to be current smokers, consume approximately 1 alcoholic drink/day, and to be socially isolated, but were slightly less likely to be depressed than those who reported minimal levels of coffee intake (<1 cup/day; see Table 1). The distribution of sociodemographic factors by coffee consumption was comparable for both the happiness and optimism analytic samples.

On average, most participants reported fairly elevated levels of psychological well-being over the follow-up period. With respect to happiness, 71.6% of women reported sustained high levels, and among those, most (89.9%) did so in two consecutive assessments. Fewer (31.4%) reported sustained high optimism, but again, most of those women (82.6%) reported high levels in two consecutive assessments. Levels of within-subject variance in psychological well-being were low over the follow-up period, indicating that happiness and optimism were highly stable over their respective three follow-up assessments ($CV_{happiness}$ = 0.15, 95% CI = 0.15–0.15; $CV_{optimism}$ = 0.11, 95% CI = 0.11–0.11). As for coffee consumption, in the happiness and optimism analytic samples, 55.0% and 36.9% of women reported sustained coffee intake of 1–3 cups/day, respectively, with the vast majority (91.6% and 100%, respectively) doing so in two consecutive assessments.

## Baseline coffee consumption and sustained high psychological well-being

Associations between baseline coffee consumption and sustained high psychological well-being are provided in Table 2. In minimally adjusted models, heavy coffee drinkers

**Table 1. Age-standardized characteristics of women across levels of caffeinated coffee consumption at each analytic baseline.**

| | 1990 Baseline (N = 44,449) | | | 2002 Baseline (N = 36,729) | | |
|---|---|---|---|---|---|---|
| | <1 cup/day (n = 18,564) | 1–3 cups/day (n = 20,635) | ≥4 cups/day (n = 5,250) | <1 cup/ day (n = 17,823) | 1–3 cups/day (n = 17,149) | ≥4 cups/day (n = 1,757) |
| | Percent or Mean (SD) | Percent or Mean (SD) | Percent or Mean (SD) | Percent or Mean (SD) | Percent or Mean (SD) | Percent or Mean (SD) |
| Age, *Years** | 56.0 (7.2) | 56.2 (7.0) | 55.0 (6.8) | 66.8 (6.9) | 66.5 (6.7) | 65.1 (6.4) |
| White, % | 97.8 | 98.5 | 99.0 | 97.7 | 98.6 | 98.9 |
| Highest education level obtained as RN degree, % | 69.0 | 67.6 | 71.0 | 67.7 | 66.9 | 70.6 |
| Married or in a relationship, % | 84.6 | 83.2 | 79.4 | 78.3 | 76.8 | 69.1 |
| BMI, *kg/m²* | 25.4 (4.8) | 25.0 (4.4) | 24.8 (4.3) | 26.5 (5.1) | 26.1 (4.7) | 26.0 (4.8) |
| Current smoker, % | 10.5 | 17.2 | 36.4 | 4.7 | 9.2 | 26.1 |
| Smoking, *Total cigarettes/day* | 13.8 (9.8) | 13.0 (9.4) | 16.1 (10.1) | 10.3 (8.2) | 10.4 (7.9) | 12.0 (7.9) |
| Physical activity, *MET-hours/week* | 2.4 (3.8) | 2.3 (3.7) | 2.0 (3.5) | 2.5 (3.4) | 2.6 (3.5) | 2.4 (3.5) |
| Average alcohol intake of 1 drink/ day, % | 14.9 | 24.1 | 21.8 | 17.7 | 27.1 | 20.8 |
| Diet quality, *Total Diet Score* | 53.0 (11.2) | 52.8 (10.9) | 51.4 (11.0) | 56.4 (12.3) | 55.9 (12.0) | 54.2 (11.9) |
| Social integration levels, % | | | | | | |
| Highly socially isolated | 10.7 | 13.0 | 15.8 | 9.6 | 11.9 | 15.0 |
| Moderately socially isolated | 24.9 | 27.7 | 30.0 | 22.6 | 24.7 | 25.5 |
| Moderately socially integrated | 34.5 | 34.9 | 33.1 | 36.7 | 36.7 | 37.4 |
| Highly socially integrated | 29.9 | 24.5 | 21.1 | 31.1 | 26.7 | 22.0 |
| Depressed, % | 8.2 | 7.2 | 6.3 | 11.5 | 10.0 | 9.6 |

[a] Values are means (standard deviation [SD]) or medians (Q25, Q75) for continuous variables and percentages for categorical variables. All values are standardized to the age distribution of the study population.

[b] Values of polytomous variables may not sum to 100% due to rounding.

[c] Not age adjusted

**Table 2. Generalized estimating equations with a Poisson distribution evaluating the association of baseline coffee intake with likelihood of reporting sustained high levels of happiness (N = 44,449) and optimism (N = 36,729)[a,b].**

| | Coffee Intake Levels | Model 1 | Model 2 | Model 3 | Model 4 |
|---|---|---|---|---|---|
| | n (%) | RR (95% CI) | RR (95% CI) | RR (95% CI) | RR (95% CI) |
| | | Sustained High Happiness Between 1992 and 2000 | | | |
| **Coffee Intake in 1990** | | | | | |
| Less than 1 cup/day | 18,564 (41.8) | Reference | Reference | Reference | Reference |
| 1–3 cups/day | 20,635 (46.4) | 1.00 (0.98–1.01) | 1.00 (0.98–1.01) | 1.00 (0.99–1.02) | 1.00 (0.99–1.01) |
| 4 or more cups/day | 5,250 (11.8) | 0.96 (0.94–0.98)**** | 0.97 (0.95–0.99)** | 0.98 (0.96–1.00) | 0.97 (0.95–0.99)** |
| *Linear p-trend* | | | | | 0.24 |
| Frequency (continuous, per 1-SD) | | 0.99 (0.99–1.00)** | 0.99 (0.99–1.00)* | 1.00 (0.99–1.00) | 1.00 (0.99–1.00) |
| | | Sustained High Optimism Between 2004 and 2012 | | | |
| **Coffee Intake in 2002** | | | | | |
| Less than 1 cup/day | 17,823 (48.5) | Reference | Reference | Reference | Reference |
| 1–3 cups/day | 17,149 (46.7) | 1.03 (1.00–1.06) | 1.02 (0.99–1.05) | 1.04 (1.01–1.07)* | 1.03 (1.00–1.06)* |
| 4 or more cups/day | 1,757 (4.8) | 1.00 (0.93–1.07) | 1.03 (0.96–1.10) | 1.05 (0.98–1.12) | 1.03 (0.96–1.11) |
| *Linear p-trend* | | | | | 0.04 |
| Frequency (continuous, per 1-SD) | | 1.01 (0.99–1.02) | 1.01 (0.99–1.02) | 1.01 (1.00–1.03) | 1.01 (1.00–1.03) |

[a] Sociodemographic covariates included in Model 1 include age, race, education, and marital status. Model 2 additionally includes information on health behaviors compiled into a single lifestyle score that includes BMI, diet quality [AHEI], alcohol consumption, physical activity and smoking. Model 3 further adjusts for social integration. Fully adjusted Model 4 includes covariates from all prior models, plus depression defined using physician diagnoses and antidepressant use.

[b] CI = confidence interval; RR = relative risk; SD = standard deviation.

****$p \leq 0.0001$;

***$p \leq 0.001$;

**$p \leq 0.01$;

*$p \leq 0.05$

experienced a 4% reduced likelihood of sustained happiness over time (RR$_{\geq 4cups/day}$ = 0.96; 95% CI = 0.94–0.98) compared to those who consumed <1 cup/day of coffee, whereas drinking 1–3 cups/day was unrelated to happiness. The magnitude of effect estimates remained similar after adjustment for behavioral, social, and psychological factors (e.g., fully adjusted RR$_{\geq 4cups/day}$ = 0.97; 95% CI = 0.95–0.99; linear *p*-trend = 0.24). In contrast, moderate coffee drinking was associated with a slightly increased likelihood of sustained optimism (minimally adjusted RR$_{1-3cups/day}$ = 1.03, 95% CI = 1.00–1.06; fully adjusted RR$_{1-3cups/day}$ = 1.03, 95% CI = 1.00–1.06). With regards to heavy coffee consumption (i.e., $\geq 4$ cups/day), effect estimates were similar in magnitude but less precise as reflected by wider confidence intervals (e.g., fully adjusted RR = 1.03, 95% CI = 0.96–1.11). Associations between continuous standardized coffee intake frequency and sustained psychological well-being levels were null for both happiness and optimism in fully adjusted models.

## Baseline psychological well-being and sustained moderate coffee consumption

As shown in Table 3, across all models, moderate and high levels (versus low levels) of happiness were not associated with sustained coffee consumption of 1–3 cups/day. High initial optimism levels, however, were modestly associated with a greater likelihood of sustaining moderate level of coffee consumption over the study period. For instance, compared to women with low optimism levels, those with moderate or high levels were 4 to 6% more likely to consistently drink 1–3 cups of coffee/day, respectively (fully adjusted models: RR$_{moderate}$ = 1.04, 95% CI = 1.00–1.08;

**Table 3. Generalized estimating equations with a Poisson distribution evaluating the association of baseline levels of happiness (N = 43,489) and optimism (N = 31,441) respectively with likelihood of reporting sustained moderate caffeinated coffee intake[a,b].**

| | Psychological Well-Being Levels | Model 1 | Model 2 | Model 3 | Model 4 |
|---|---|---|---|---|---|
| | n (%) | RR (95% CI) | RR (95% CI) | RR (95% CI) | RR (95% CI) |
| | | Sustained Moderate Coffee Intake 1994 and 2010 | | | |
| **Happiness in 1992** | | | | | |
| Low | 6,246 (14.4) | Reference | Reference | Reference | Reference |
| Moderate | 8,103 (18.6) | 1.00 (0.97–1.03) | 1.00 (0.97–1.03) | 1.01 (0.98–1.04) | 1.00 (0.97–1.03) |
| High | 29,140 (67.0) | 1.01 (0.98–1.03) | 1.00 (0.97–1.02) | 1.02 (0.99–1.04) | 1.01 (0.99–1.04) |
| *Linear p-trend* | | | | | 0.34 |
| Frequency (continuous, per 1-SD) | | 1.00 (0.99–1.01) | 1.00 (0.99–1.01) | 1.00 (0.99–1.01) | 1.00 (0.99–1.01) |
| | | Sustained Moderate Coffee Intake 2006 and 2010 | | | |
| **Optimism in 2004** | | | | | |
| Low | 9,867 (31.4) | Reference | Reference | Reference | Reference |
| Moderate | 9,659 (30.7) | 1.03 (1.00–1.07) | 1.03 (0.99–1.07) | 1.04 (1.00–1.08)* | 1.04 (1.00–1.08) |
| High | 11,915 (37.9) | 1.05 (1.01–1.09)** | 1.04 (1.00–1.08)* | 1.06 (1.02–1.10)*** | 1.06 (1.02–1.10)** |
| *Linear p-trend* | | | | | 0.05 |
| Frequency (continuous, per 1-SD) | | 1.03 (1.01–1.04)*** | 1.02 (1.01–1.04)** | 1.03 (1.02–1.05)**** | 1.03 (1.01–1.05)*** |

[a] Sociodemographic covariates included in Model 1 include age, race, education, and marital status. Model 2 additionally includes information on health behaviors compiled into a single lifestyle score that includes BMI, diet quality [AHEI], alcohol consumption, physical activity and smoking. Model 3 further adjusts for social integration. Fully adjusted Model 4 includes covariates from all prior models, plus depression defined using physician diagnoses and antidepressant use.

[b] CI = confidence interval; RR = relative risk; SD = standard deviation.

****$p \leq 0.0001$;

***$p \leq 0.001$;

**$p \leq 0.01$;

*$p \leq 0.05$

$RR_{high}$ = 1.06, 95% CI = 1.02–1.10; linear $p$-trend = 0.05). Similarly, higher continuous optimism but not happiness scores were modestly related to greater likelihood of consistently drinking 1–3 cups of coffee/day over time (e.g., fully adjusted $RR_{per\ 1\text{-}SD\ optimism}$ = 1.03, 95% CI = 1.01–1.05; fully adjusted $RR_{per\ 1\text{-}SD\ happiness}$ = 1.00, 95% CI = 0.99–1.01).

## Sensitivity analyses

Results from sensitivity analyses with other beverages and total caffeine intake are provided S3 and S4 Tables in S1 File. Regardless of consumption levels, prospective associations of decaffeinated coffee and caffeinated tea with sustained happiness or optimism, respectively, were generally null. Associations of total caffeine consumption mirrored the main results: the likelihood of maintaining high happiness levels was lower in women with heavier caffeine intake (e.g., fully adjusted $RR_{\geq 550g/day}$ = 0.97, 95% CI = 0.95–0.99), whereas the likelihood of sustaining high optimism levels appeared somewhat higher in women with moderate levels of caffeine intake (e.g., fully adjusted $RR_{100\text{-}549g/day}$ = 1.03, 95% CI = 1.00–1.06).

Accounting for baseline outcome measures in both the primary and secondary models attenuated most effect estimates (see S5 Table in S1 File). For instance, after controlling for initial happiness levels, heavier coffee intake was no longer associated with a reduced likelihood of sustained high happiness over the follow-up period (e.g., fully adjusted $RR_{\geq 4cups/day}$ = 0.98, 95% CI = 0.96–1.01). Similarly, after adjusting for initial optimism levels, all associations became null; although the magnitude of effect estimates was consistent with results from the primary analyses, the confidence intervals were wider (e.g., $RR_{1\text{-}3cups/day}$ = 1.02, 95%

CI = 0.98–1.06). After adjusting for baseline coffee consumption levels, high versus low levels of happiness at the study onset were modestly associated with increased likelihood of maintaining moderate levels of coffee consumption over the follow-up period (e.g., fully adjusted $RR_{high}$ = 1.03, 95% CI = 1.00–1.05). Conversely, after adjusting for baseline coffee consumption, associations between baseline optimism and sustained moderate coffee intake over time became weaker in all models (e.g., fully adjusted $RR_{high}$ = 1.03, 95% CI = 0.99–1.07). Across models, all effect sizes were fairly small.

## Discussion

In light of prior work indicating that coffee consumption was related to a reduced risk of suicide and depression [13, 14], this study sought to determine whether coffee intake was associated with a greater likelihood of sustaining high levels of psychological well-being over time. Results suggested that initial levels of coffee intake were weakly (if at all) associated with maintaining happiness and optimism over 10 years of follow-up, and the associations were relatively unstable across different models and sensitivity analyses with decaffeinated coffee, tea, and total caffeine intake. When evaluating bidirectionality, secondary analyses testing psychological well-being's role in subsequent coffee intake similarly showed weak and inconsistent associations, particularly when adjusting for baseline outcome levels. The absence of a meaningful relationship is noteworthy because we were able to examine these associations within a large sample, which should increase our ability to detect small but stable effects. Additionally, several prospective studies–including previous investigations conducted in the NHS cohort–have identified a robust relationship between coffee consumption and a reduced risk of both suicide [12, 13] and incident depression [11, 14]. Taken together, these findings show that while coffee consumption may mitigate depression and suicide risk among middle-age and older women, it does not appear to be appreciably related to psychological well-being over time.

These results are markedly different from previously documented associations between psychological well-being and other health behaviors, including physical activity, diet, and tobacco use [6, 7, 37]. Although most prior work has sought to determine whether psychological well-being is related to future engagement in healthy behaviors, recent evidence finds that that this relationship is likely bidirectional, with healthier behaviors also contributing to higher levels of psychological well-being over time [6, 7]. Like other healthy lifestyle behaviors, coffee consumption is associated with a reduced risk of morbidity and mortality across a range of health conditions [8]. Therefore, the failure to find evidence of similar bidirectional associations with psychological well-being is notable. Health-enhancing behaviors–including coffee intake–are posited to influence psychological states in part through their impact on neurophysiologic pathways that underlie mood (e.g., physical activity stimulates serotonin and endorphin production [38], coffee intake increases dopamine signaling in the brain [39]). Potential reverse associations, however, are typically attributed to psychological well-being's role as a resource that can help individuals sustain healthy behaviors in the face of challenging environmental conditions [6, 40]. Barriers to coffee drinking, however, are relatively few as it is a socially normative behavior that is widely available and easily accessible; moreover, it has some addictive properties due to its caffeine content [41, 42]. As a result, it is plausible that the benefits of psychological well-being related to more effective regulation of emotions and behavior [6] may be less necessary to support the maintenance of coffee consumption habits compared to traditional health behaviors, which require more effortful action.

Although observed associations between coffee intake and psychological well-being were not appreciable, some small differences were evident. Given the large sample sizes used in the present analyses, this study was highly powered to detect even minor differences between

women of varying levels of coffee consumption, perhaps resulting in the identification of associations with limited clinical relevance. However, given that most positive associations were found with caffeinated coffee and total caffeine intake, it is also possible that modest effect estimates reflect true incremental differences in the likelihood of sustaining psychological well-being that may be attributable to shared biological mechanisms linking coffee intake and reduced depression risk [43] (e.g., caffeine's promotion of dopamine signaling [39]). Physiologically plausible pathways linking coffee intake and psychological well-being may warrant further examination in randomized intervention studies before completely discarding coffee consumption as factor that might influence and be influenced by psychological well-being.

This study has some limitations. Single-item measures of psychological well-being tend to be more readily available in large-scale epidemiologic studies, which typically include few (if any) positive psychological indicators; however, a single-item measure used to quantify happiness may not adequately capture the full extent of the construct. This could partly account for the lack of stability in our findings. Another limitation relates to the sample which was comprised of older, predominantly White health professional women. Although geographically varied, participants were not representative of racial/ethnic or socioeconomically diverse populations. Relatedly, due to their occupation, NHS women may be particularly health-conscious, which could attenuate associations otherwise existing in a more varied sample of occupations. Lastly, our findings are based on observational data, which has the usual limitations regarding the ability to infer causality between predictors and outcomes.

This study also has a number of strengths. NHS is a large prospective cohort study with an extensive array of rich data obtained over multiple decades. Information on two distinct dimensions of psychological well-being were collected, which enabled the evaluation of whether associations were unique to a single dimension or more broadly observed. Nuanced measures of coffee intake and related constructs were also collected at multiple time points over a 20-year period. Available data allowed for both the testing of potential bidirectional relationships and several sensitivity analyses using data on related beverages (i.e., decaffeinated coffee, caffeinated tea, total caffeine intake). Lastly, analyses statistically controlled for a comprehensive set of potential confounders, including psychosocial factors like depression and social integration.

## Conclusion

Identifying the causes and consequences of psychological well-being is a growing public health priority, due not only to its promise as a novel intervention strategy to increase likelihood of healthy aging in the population [6, 44–46], but also to the value of increasing individuals' sense of well-being as its own end [1]. Prospective studies have found associations between coffee intake and a reduced risk of depression [14] and suicide [13], as well as between psychological well-being and the adoption of healthy behaviors over time [6, 7]. However, the current study did not find substantive associations between coffee intake and psychological well-being over up to 20 years of follow-up in a large-scale cohort of midlife and older women. Although our null findings demonstrate that coffee drinking is largely unrelated to psychological well-being, it is noteworthy that coffee consumption did not confer any apparent harm. Maintaining moderate levels of coffee consumption may be a promising approach to promote other dimensions of health with limited impact on psychological well-being over time.

## Supporting information

**S1 File.**
(DOCX)

## Acknowledgments

We would like to thank the participants and the staff of the Nurses' Health Study for their valuable contributions. The authors assume full responsibility for the analyses and interpretation of these data.

## Author Contributions

**Conceptualization:** Farah Qureshi, Meir Stampfer, Laura D. Kubzansky, Claudia Trudel-Fitzgerald.

**Data curation:** Meir Stampfer.

**Formal analysis:** Claudia Trudel-Fitzgerald.

**Funding acquisition:** Meir Stampfer.

**Methodology:** Farah Qureshi, Meir Stampfer, Laura D. Kubzansky, Claudia Trudel-Fitzgerald.

**Project administration:** Farah Qureshi, Claudia Trudel-Fitzgerald.

**Writing – original draft:** Farah Qureshi.

**Writing – review & editing:** Farah Qureshi, Meir Stampfer, Laura D. Kubzansky, Claudia Trudel-Fitzgerald.

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
