## [Decision Letter · Decision Letter 0]

11 Apr 2022

Prospective associations between coffee consumption and psychological well-being

PONE-D-21-29419

Dear Dr. Trudel-Fitzgerald,

We’re pleased to inform you that your manuscript has been judged scientifically suitable for publication and will be formally accepted for publication once it meets all outstanding technical requirements.

Kind regards,

Masaki Mogi

Academic Editor

PLOS ONE

Additional Editor Comments (optional):

Reviewers' comments:

Reviewer's Responses to Questions

**Comments to the Author**

1. Is the manuscript technically sound, and do the data support the conclusions?

Reviewer #1: Yes

Reviewer #2: Yes

2. Has the statistical analysis been performed appropriately and rigorously? 

Reviewer #1: Yes

Reviewer #2: Yes

3. Have the authors made all data underlying the findings in their manuscript fully available?

Reviewer #1: Yes

Reviewer #2: No

4. Is the manuscript presented in an intelligible fashion and written in standard English?

Reviewer #1: Yes

Reviewer #2: Yes

5. Review Comments to the Author

Reviewer #1: Based on a large sample from the Nurses Health Study cohort, the authors examined in the present study whether coffee consumption might be related to happiness, positive mood and hence better well-being. This research hypothesis was undertaken on the basis of the fact that coffee consumption alleviates some depression symptoms and reduces risk of depression. Surprisingly, at the other end of the spectrum, happiness and good mood, coffee consumption has almost no effect.

The study was well-designed and carefully performed with a thorough analysis grouping several models and the results were homogeneous. I really congratulate the authors for their study which was truly needed in light of what we know about the relationship between depression and suicide. This study will prevent people to extrapolate from the depression end to the effects of coffee consumption in individuals with normal levels of mood. The present manuscript is well written, easy to read and does not need any change in my eyes.

Reviewer #2: This is a comprehensive analysis of the association between coffee consumption and optimism as well as happiness, with both outcomes approached by self-reported participant’s information using well-recognized tests. Although previous work showed an inverse association between coffee and lower risk of depression and suicide, the authors were not able to find an association for the studied outcomes.

Strengths of this work include an exhaustive information about Methods, a very careful use of adjustment for potential confounders, and the use of a large population from a well characterized cohort, with prolonged follow-up.

6. PLOS authors have the option to publish the peer review history of their article (what does this mean?). If published, this will include your full peer review and any attached files.

Reviewer #1: No

Reviewer #2: No

---

## [Editor Report · Acceptance letter]

25 Apr 2022

PONE-D-21-29419 

Prospective associations between coffee consumption and psychological well-beingProspective associations between coffee consumption and psychological well-being 

Dear Dr. Trudel-Fitzgerald:

I'm pleased to inform you that your manuscript has been deemed suitable for publication in PLOS ONE. Congratulations! Your manuscript is now with our production department. 

Kind regards, 

on behalf of

Dr. Masaki Mogi 

Academic Editor

PLOS ONE